# *Walk These Ways*: Tuning Robot Control for Generalization with Multiplicity of Behavior

**Gabriel B. Margolis**     **Pulkit Agrawal**

Improbable AI Lab, Massachusetts Institute of Technology

{gmargo, pulkitag}@mit.edu

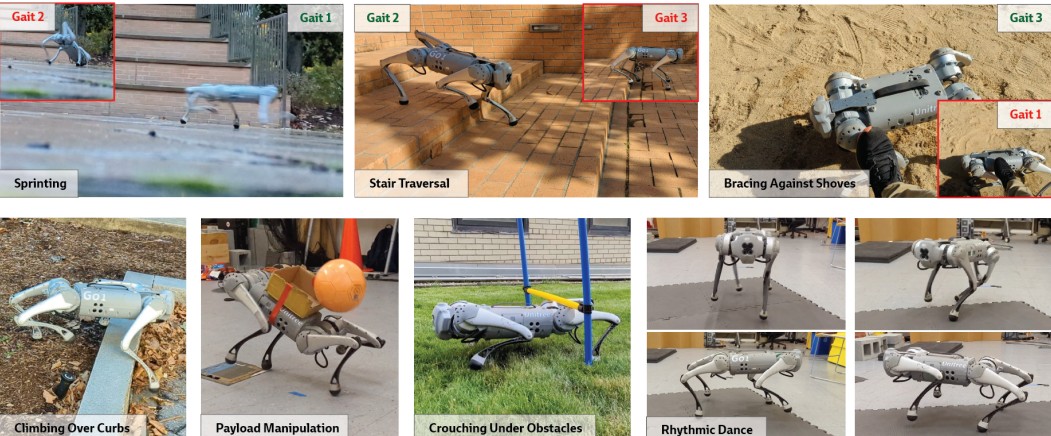

Figure 1: Multiplicity of Behavior (MoB) enables a human to tune a *single* quadruped policy trained *on flat ground* to diverse unseen environments. **Top row:** A low-frequency gait fails to sprint on slippery terrain (Gait 2; inset) but tuning it to high frequency results in success (Gait 1). However, a low frequency and high footswing height are necessary for stair traversal (Gait 2; middle image). A low footswing and wide stance (Gait 3) makes the robot robust to leg shoves, but Gait 1, which succeeded at sprinting, fails. Tuning gait thus aids in generalizing to different tasks. **Bottom row:** Examples of other behaviors enabled by our controller.

**Abstract:** Learned locomotion policies can rapidly adapt to diverse environments similar to those experienced during training but lack a mechanism for fast tuning when they fail in an out-of-distribution test environment. This necessitates a slow and iterative cycle of reward and environment redesign to achieve good performance on a new task. As an alternative, we propose learning a single policy that encodes a structured family of locomotion strategies that solve training tasks in different ways, resulting in *Multiplicity of Behavior* (MoB). Different strategies generalize differently and can be chosen in real-time for new tasks or environments, bypassing the need for time-consuming retraining. We release a fast, robust open-source MoB locomotion controller, *Walk These Ways*, that can execute diverse gaits with variable footswing, posture, and speed, unlocking diverse downstream tasks: crouching, hopping, high-speed running, stair traversal, bracing against shoves, rhythmic dance, and more. Video and code release: https://gmargo11.github.io/walk-these-ways/

**Keywords:** Locomotion, Reinforcement Learning, Task Specification

## 1   Introduction

Recent works have established that quadruped locomotion controllers trained with reinforcement learning in simulation can successfully be transferred to traverse challenging natural terrains [1, 2,

6th Conference on Robot Learning (CoRL 2022), Auckland, New Zealand.

3]. Adaptation to diverse terrains is accomplished by estimating terrain properties from sensory observations that are then used by the controller (i.e., *online system identification*). The success of this paradigm relies on two assumptions: a priori modeling of environment parameters that can vary during deployment and the ability to estimate these parameters from sensory observations. To bypass the first assumption, one possibility is to widely vary a large set of environment parameters during training. However, this creates a hard learning problem due to creation of challenging or infeasible locomotion scenarios. To simplify learning, typically the designer chooses a subset of parameters that are randomized in a carefully restricted range. Even in this setup, additional measures such as a learning curriculum and reward shaping are necessary for successful learning in simulation.

As a result of these practical restrictions on the expressiveness of the simulation, quite often the robot encounters scenarios during deployment that were not modeled during training. For instance, if the robot is only presented with flat ground and terrain geometry is not varied during training, it may fail to traverse non-flat terrains such as stairs. In such a case, it is common to tweak the training environments or the reward functions and re-train the policy. This iterative loop of re-training and real-world testing is tedious. To make things worse, in some scenarios such iteration is insufficient because it is not possible to accurately model or sense important environment properties. For example, thick bushes are both hard to simulate due to compliance and hard to sense because depth sensors do not distinguish them from walls. Thus, the robot may attempt to climb over thick bushes like a rock or move through them with an overly conservative gait that leaves the robot stuck.

The examples above illustrate that even for the most advanced *sim-to-real* systems, the real world offers new challenges. We broadly refer to scenarios that can be simulated but are not anticipated during training and the situations which cannot be simulated or identified from sensory observations as *out-of-distribution* cases. We present a framework for policy learning that enables improved performance in *out-of-distribution* scenarios under some assumptions detailed below. Our key insight is that given a task, there are multiple equally good solutions (i.e., under-specification [4]) that have equivalent training performance but can generalize in different ways. For instance, the task of walking on flat ground only imposes a constraint on the velocity of robot's body, but not on how the legs should move, or high should the torso be above the ground, etc. Consider two different walking behaviors: *crouch* where the robot keeps its torso close to the ground and *stomp* where the torso is high and also the legs have a high foot swing. While both *crouch* and *stomp* succeed at walking on flat ground, their generalization to *out-of-distribution* scenarios is different: with *crouch* the robot can traverse under obstacles but not stairs, whereas with *stomp* it can climb over curbs/stairs but not move under obstacles (Figure 1).

Out of the many possible locomotion behaviors that succeed in the training environment, typical reinforcement learning formulations result in a policy that only embodies one solution and therefore expresses a single generalizing bias. To facilitate generalization to diverse scenarios, we propose a technique, *Multiplicity of Behavior* (MoB), that given the same observation history and a small set of *behavior parameters* outputs different walking behaviors. When faced with an unseen scenario, one can test different behaviors by varying these parameters, which affords much quicker iteration than re-training a new policy and facilitates collection of online demonstrations by a human pilot.

The utility of MoB depends on the assumption that some subset of behaviors successful in the restricted training environment will also succeed in the out-of-distribution target environment. To demonstrate that this is true in a meaningful sense, we chose an extreme case: train a *single policy* for quadrupedal walking only on *flat ground* and evaluate on non-flat terrains and new tasks. We show that a human operator can tune behaviors in real-time to enable successful locomotion in the presence of diverse unseen terrains and dynamics including uneven ground, stairs, external shoves, and constrained spaces (Figure 1). The same tuning mechanisms can be used to compose behaviors to perform new tasks such as payload manipulation and rhythmic dance (Figure 1).

This work contributes a robust low-level quadruped controller that can execute diverse structured behaviors, which we we hope will be a useful building block for future locomotion applications. It can serve as a platform for collecting quadruped demonstrations for diverse tasks which is enabled by an interpretable high-level control interface. Furthermore, our controller showcases MoB as a practical tool for out-of-distribution generalization. While our implementation of MoB leverages expert knowledge in the domain of locomotion, in general, the technique of learning multiple methods of achieving goals to facilitate generalization is a promising approach with potential for broad application in sim-to-real reinforcement learning for robotics.

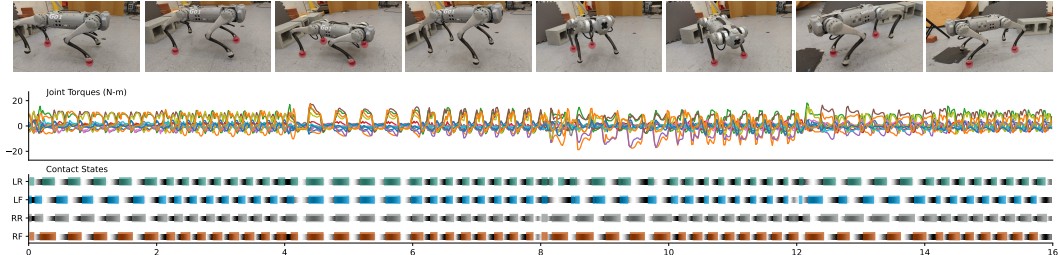

Figure 2: A learned controller transitions between classical structured gaits: trotting, pronking, pacing, and bounding in place at alternating frequencies 2Hz and 4Hz. Images show the robot achieving contact phases of each gait, with stance feet highlighted in red. Black shading in the bottom plot reflects the timing reference variables $\mathbf{t}_t$ for each foot; colored bars report the contact states measured by foot sensors. As we later expose, diverse behaviors can facilitate novel downstream capabilities.

## 2   Background

**Auxiliary Rewards.** Locomotion gaits learned using only the task reward (e.g. velocity tracking) have not been shown to successfully transfer to the real world. It is necessary to train using auxiliary rewards that bias the robot to maintain a particular contact schedule, action smoothness, energy consumption, or foot clearance [1, 2, 3, 5, 6, 7], to compensate for the sim-to-real gap. Such auxiliary rewards can be interpreted as biases for generalization. For example, foot clearance enables the robot to be robust when the terrain in the real world is more uneven, or the robot's body sinks more in the real world than in simulation [7]. If an agent fails on a real-world task, it is common practice to manually tune auxiliary rewards to encourage the emergence of successful real-world behavior. However, such tuning is tedious because it requires repeated iterations of training and deployment. Furthermore, such tuning is task-specific and must be repeated for new tasks commanded to the agent. The difficulty of designing a single set of auxiliary rewards that promote generalization in diverse set of downstream tasks is illustrated in the top row insets of Figure 1: each shows an instance where a fixed auxiliary reward would lead to failure for one task, despite working well for another.

**Learning Locomotion with Parameterized Style.** Closely related recent work on bipeds has explicitly included gait parameters in the task specification through parameter-augmented auxiliary reward terms, including tracking rewards for contact patterns [8] and target foot placements [9]. [8] used gait-dependent reward terms in bipedal locomotion to control the timing offset between the two feet and the duration of the swing phase. The reward structure of [8] inspired ours, but due to its small space of command parameters, did not propose and would not support application to compositional tasks or out-of-distribution environments. To similar effect, other works have imposed parameterized style through libraries of reference gaits. [10, 11] generated a library of reference trajectories and trained a goal-conditioned policy to imitate them. [12] demonstrated that a small discrete set of motion styles can be learned simultaneously from a reference trajectory library.

**Learning with Diversity Objectives.** Several prior methods aim to automatically learn a collection of high-performing and diverse behaviors. Quality diversity (QD) methods [13, 14, 15] learn a library of diverse policies by enforcing a novelty objective defined among trajectories. They typically perform this optimization using evolutionary strategies. QD has demonstrated benefits including improved optimization performance and reuse of skills for online adaptation [13, 14]. Another line of work uses unsupervised objectives for skill discovery in RL, towards improving optimization [16] or out-of-distribution generalization [17, 18]. Unsupervised diversity approaches hold promise, but they have not yet scaled to real robotic platforms and do not produce an grounded interface of parameters for guiding behavior.

**Hierarchical Control Using Gait Parameters.** Several works have learned a high-level policy to accomplish downstream tasks by modulating the gait parameters of a low-level model-based controller. This approach has been previously applied to energy minimization [19] and vision-guided foot placement [20, 21]. These works relied on a model-predictive low-level controller to execute the different gaits in absence of a gait-conditioned learned controller. The tradeoffs between model-based control and reinforcement learning are well discussed in prior literature [3, 22]. Our work enables revisiting hierarchical approaches with learning at both the high and low levels.

| Term | Equation | Weight |
|---|---|---|
| $r_{v_{x,y}^{\text{cmd}}}$: xy velocity tracking | $\exp\{-\lvert\mathbf{v}_{xy}-\mathbf{v}_{xy}^{\text{cmd}}\rvert^2/\sigma_{vxy}\}$ | 0.02 |
| $r_{\omega_z^{\text{cmd}}}$: yaw velocity tracking | $\exp\{-(\boldsymbol{\omega}_z-\boldsymbol{\omega}_z^{\text{cmd}})^2/\sigma_{\omega z}\}$ | 0.01 |
| $r_{c_f^{\text{cmd}}}$: swing phase tracking (force) | $\sum_{\text{foot}}[1-C_{\text{foot}}^{\text{cmd}}(\boldsymbol{\theta}^{\text{cmd}},t)]\exp\{-\lvert\mathbf{f}^{\text{foot}}\rvert^2/\sigma_{cf}\}$ | $-0.08$ |
| $r_{c_v^{\text{cmd}}}$: stance phase tracking (velocity) | $\sum_{\text{foot}}[C_{\text{foot}}^{\text{cmd}}(\boldsymbol{\theta}^{\text{cmd}},t)]\exp\{-\lvert\mathbf{v}_{xy}^{\text{foot}}\rvert^2/\sigma_{cv}\}$ | $-0.08$ |
| $r_{h_z^{\text{cmd}}}$: body height tracking | $(\boldsymbol{h_z}-\boldsymbol{h_z^{\text{cmd}}})^2$ | $-0.2$ |
| $r_{\phi^{\text{cmd}}}$: body pitch tracking | $(\boldsymbol{\phi}-\boldsymbol{\phi}^{\text{cmd}})^2$ | $-0.1$ |
| $r_{s_y^{\text{cmd}}}$: raibert heuristic footswing tracking | $(\mathbf{p}_{x,y\,\text{foot}}^f-\mathbf{p}_{x,y\,\text{foot}}^{f\;\text{cmd}}(\boldsymbol{s}_y^{\text{cmd}}))^2$ | $-0.2$ |
| $r_{h_z^{f\,\text{cmd}}}$: footswing height tracking | $\sum_{\text{foot}}(\boldsymbol{h}_{z\,\text{foot}}^f-\boldsymbol{h}_z^{f\;\text{cmd}})^2 C_{\text{foot}}^{\text{cmd}}(\boldsymbol{\theta}^{\text{cmd}},t)$ | $-0.6$ |
| z velocity | $\mathbf{v}_z^2$ | $-4\text{e-}4$ |
| roll-pitch velocity | $\lvert\boldsymbol{\omega}_{xy}\rvert^2$ | $-2\text{e-}5$ |
| foot slip | $\lvert\mathbf{v}_{xy}^{\text{foot}}\rvert^2$ | $-8\text{e-}4$ |
| thigh/calf collision | $\mathbb{1}_{\text{collision}}$ | $-0.02$ |
| joint limit violation | $\mathbb{1}_{q_i>q_{max}\lvert\lvert q_i<q_{min}}$ | $-0.2$ |
| joint torques | $\lvert\boldsymbol{\tau}\rvert^2$ | $-2\text{e-}5$ |
| joint velocities | $\lvert\dot{\mathbf{q}}\rvert^2$ | $-2\text{e-}5$ |
| joint accelerations | $\lvert\ddot{\mathbf{q}}\rvert^2$ | $-5\text{e-}9$ |
| action smoothing | $\lvert\mathbf{a}_{t-1}-\mathbf{a}_t\rvert^2$ | $-2\text{e-}3$ |
| action smoothing, 2nd order | $\lvert\mathbf{a}_{t-2}-2\mathbf{a}_{t-1}+\mathbf{a}_t\rvert^2$ | $-2\text{e-}3$ |

*(right-side annotations: Task, Augmented Auxiliary, Fixed Auxiliary)*

Table 1: Reward structure: task rewards, augmented auxiliary rewards, and fixed auxiliary rewards.

## 3 Method

To obtain MoB, we train a conditional policy $\pi(\cdot\lvert\mathbf{c}_t,\mathbf{b}_t)$ that achieves tasks specified by the command ($\mathbf{c}_t$) in *multiple* ways that result from different choices of behavior parameters, $\mathbf{b}_t$. The question arises of how to define $\mathbf{b}_t$. We could learn behaviors using an unsupervised diversity metric, but these behaviors might not be useful [4] and are not human tunable. To overcome these issues, we leverage human intuition about useful behavior parameters ($\mathbf{b}_t$) corresponding to gait properties like foot swing motion, body posture, and contact schedule [6, 7, 8, 23]. During training, the agent receives a combination of task rewards (for velocity tracking), fixed auxiliary rewards (to promote sim-to-real transfer and stable motion), and finally augmented auxiliary rewards (that encourage locomotion in the desired style). During deployment in a novel environment, a human operator can tune behavior of the policy by changing its input $\mathbf{b}_t$.

### 3.1 Task Structure for MoB

**Task Specification.** We consider the task of omnidirectional velocity tracking. This task is specified by a 3-dimensional command vector $\mathbf{c}_t=[\mathbf{v}_x^{\text{cmd}},\mathbf{v}_y^{\text{cmd}},\boldsymbol{\omega}_z^{\text{cmd}}]$ where $\mathbf{v}_x^{\text{cmd}},\mathbf{v}_y^{\text{cmd}}$ are the desired linear velocities in the body-frame x- and y- axes, and $\boldsymbol{\omega}_z^{\text{cmd}}$ is the desired angular velocity in the yaw axis.

**Behavior Specification.** We parameterize the style of task completion by an 8-dimensional vector of behavior parameters, $\mathbf{b}_t$:

$$\mathbf{b}_t=[\boldsymbol{\theta}_1^{\text{cmd}},\boldsymbol{\theta}_2^{\text{cmd}},\boldsymbol{\theta}_3^{\text{cmd}},\boldsymbol{f}^{\text{cmd}},\boldsymbol{h}_z^{\text{cmd}},\boldsymbol{\phi}^{\text{cmd}},\boldsymbol{s}_y^{\text{cmd}},\boldsymbol{h}_z^{f\,\text{cmd}}].$$

$\boldsymbol{\theta}^{\text{cmd}}=(\boldsymbol{\theta}_1^{\text{cmd}},\boldsymbol{\theta}_2^{\text{cmd}},\boldsymbol{\theta}_3^{\text{cmd}})$ are the timing offsets between pairs of feet. These express gaits including pronking ($\boldsymbol{\theta}^{\text{cmd}}=(0.0,0,0)$), trotting ($\boldsymbol{\theta}^{\text{cmd}}=(0.5,0,0)$), bounding, ($\boldsymbol{\theta}^{\text{cmd}}=(0,0.5,0)$), pacing ($\boldsymbol{\theta}^{\text{cmd}}=(0,0,0.5)$), as well as their continuous interpolations such as galloping ($\boldsymbol{\theta}^{\text{cmd}}=(0.25,0,0)$). Taken together, the parameters $\boldsymbol{\theta}^{\text{cmd}}$ can express all two-beat quadrupedal contact patterns; Figure 2 provides a visual illustration. $\boldsymbol{f}^{\text{cmd}}$ is the stepping frequency expressed in Hz. As an example, commanding $\boldsymbol{f}^{\text{cmd}}=3\,\text{Hz}$ will result in each foot making contact three times per second. $\boldsymbol{h}_z^{\text{cmd}}$ is the body height command; $\boldsymbol{\phi}^{\text{cmd}}$ is the body pitch command. $\boldsymbol{s}_y^{\text{cmd}}$ is the foot stance width command; $\boldsymbol{h}_z^{f\,\text{cmd}}$ is the footswing height command.

**Reward function.** All reward terms are listed in Table 1. Task rewards for body velocity tracking are defined as functions of the command vector $\mathbf{c}_t$. Auxiliary rewards are used constrain the

quadruped's motion for various reasons. "Fixed" auxiliary rewards are independent of behavior parameters ($\mathbf{b}_t$) and encourage stability and smoothness across all gaits for better sim-to-real transfer. During training, one concern is that the robot might abandon its task or choose an early termination when the task reward is overwhelmed by penalties from the auxiliary objectives. To resolve this, as in [7], we force the total reward to be a positive linear function of the task reward by computing it as $r_{\text{task}} \exp\left(c_{\text{aux}} r_{\text{aux}}\right)$ where $r_{\text{task}}$ is the sum of (positive) task reward terms and $r_{\text{aux}}$ is the sum of (negative) auxiliary reward terms (we use $c_{\text{aux}} = 0.02$). This way, the agent is always rewarded for progress towards the task, more when auxiliary objectives are satisfied and less when they are not.

For MoB, we define augmented auxiliary rewards as functions of the behavior vector $\mathbf{b}_t$. We designed these rewards to increase when the realized behavior matches $\mathbf{b}_t$, and to not conflict with the task reward. This required some careful design of the reward structure. For example, when implementing stance width as a behavior parameter, a naive approach would be to simply reward a constant desired distance between left and right feet. However, this penalizes the robot during fast turning tasks requiring relative lateral motion of the feet. To avoid this, we implement the Raibert Heuristic, which suggests the necessary kinematic motion of the feet to achieve a particular body velocity and contact schedule [23, 24]. The Raibert Heuristic computes the desired foot position in the ground plane, $\boldsymbol{p}_{x,y\text{foot}}^{f\ \text{cmd}}(\boldsymbol{s}_y^{\text{cmd}})$, as an adjustment to the baseline stance width to make it consistent with the desired contact schedule and body velocity. To define the desired contact schedule, function $C_{\text{foot}}^{\text{cmd}}(\boldsymbol{\theta}^{\text{cmd}}, t)$ computes the desired contact state of each foot from the phase and timing variable, as described in [8], with details given in the appendix.

## 3.2 Learning Diversified Locomotion

**Task and Behavior Sampling** In order to learn graceful online transitions between behaviors, we resample the desired task and behavior within each training episode. To enable the robot to both run and spin fast, we sample task $\mathbf{c}_t = (\mathbf{v}_x^{\text{cmd}}, \mathbf{v}_y^{\text{cmd}}, \boldsymbol{\omega}_z^{\text{cmd}})$ using the grid adaptive curriculum strategy from [3]. Then, we need to sample a target behavior $\mathbf{b}_t$. First, we sample $(\boldsymbol{\theta}_1^{\text{cmd}}, \boldsymbol{\theta}_2^{\text{cmd}}, \boldsymbol{\theta}_3^{\text{cmd}})$ as one of the symmetric quadrupedal contact patterns (pronking, trotting, bounding, or pacing) which are known as more stable and which we found a sufficient basis for diverse useful gaits. Then, the remaining command parameters $(\mathbf{v}_y^{\text{cmd}}, \boldsymbol{f}^{\text{cmd}}, \boldsymbol{h}_z^{\text{cmd}}, \boldsymbol{\phi}^{\text{cmd}}, \boldsymbol{h}_z^{f\ \text{cmd}}, \boldsymbol{s}_y^{\text{cmd}})$ are sampled independently and uniformly. Their ranges are given in Table 6.

**Policy Input.** The input to the policy is a 30-step history of observations $\mathbf{o}_{t-H\ldots t}$, commands $\mathbf{c}_{t-H\ldots t}$, behaviors $\mathbf{b}_{t-H\ldots t}$, previous actions $\mathbf{a}_{t-H-1\ldots t-1}$, and timing reference variables $\mathbf{t}_{t-H\ldots t}$. The observation space $\mathbf{o}_t$ consists of joint positions and velocities $\mathbf{q}_t, \dot{\mathbf{q}}_t$ (measured by joint encoders) and the gravity vector in the body frame $\mathbf{g}_t$ (measured by accelerometer). The timing reference variables $\mathbf{t}_t = [\sin(2\pi t^{\text{FR}}), \sin(2\pi t^{\text{FL}}), \sin(2\pi t^{\text{RR}}), \sin(2\pi t^{\text{RL}})]$ are computed from the offset timings of each foot: $[t^{\text{FR}}, t^{\text{FL}}, t^{\text{RR}}, t^{\text{RL}}] = [t + \boldsymbol{\theta}_2^{\text{cmd}} + \boldsymbol{\theta}_3^{\text{cmd}}, t + \boldsymbol{\theta}_1^{\text{cmd}} + \boldsymbol{\theta}_3^{\text{cmd}}, t + \boldsymbol{\theta}_1^{\text{cmd}}, t + \boldsymbol{\theta}_2^{\text{cmd}}]$, where $t$ is a counter variable that advances from 0 to 1 during each gait cycle and $^{\text{FR}}, ^{\text{FL}}, ^{\text{RR}}, ^{\text{RL}}$ are the four feet. This form is adapted from [8] to express quadrupedal gaits.

**Policy Architecture.** Our policy body is an MLP with hidden layer sizes $[512, 256, 128]$ and ELU activations. Besides the above, the policy input also includes estimated domain parameters: the velocity of the robot body and the ground friction, which are predicted from the observation history using supervised learning in the manner of [7]. The estimator module is an MLP with hidden layer sizes $[256, 128]$ and ELU activations. We did not analyze the impact of this estimation on performance but found it useful for visualizing deployments.

**Action Space.** The action $\mathbf{a}_t$ consists of position targets for each of the twelve joints. A zero action corresponds to the nominal joint position, $\hat{\mathbf{q}}$. The position targets are tracked using a proportional-derivative controller with $k_p = 20, k_d = 0.5$.

## 3.3 Design Choices for Sim-to-Real Transfer

**Domain Randomization.** For better sim-to-real transfer, we train a policy that is robust to a range of robot's body mass, motor strength, joint position calibration, the ground friction and restitution, and the orientation and magnitude of gravity. As we are interested in studying out-of-distribution generalization, we only train on flat ground without any randomization of terrain geometry. This choice also simplified training. The randomization ranges of all parameters are in Appendix A.

| Gait | 0.0 m/s | 1.0 m/s | 2.0 m/s | 3.0 m/s |
|---|---|---|---|---|
| Trotting | $9_{\pm 1}$ | $24_{\pm 1}$ | $53_{\pm 5}$ | $98_{\pm 9}$ |
| Pronking | $32_{\pm 1}$ | $43_{\pm 2}$ | $68_{\pm 5}$ | $112_{\pm 5}$ |
| Pacing | $13_{\pm 3}$ | $25_{\pm 2}$ | $55_{\pm 3}$ | $99_{\pm 6}$ |
| Bounding | $22_{\pm 2}$ | $39_{\pm 4}$ | $78_{\pm 5}$ | $127_{\pm 35}$ |
| Gait-free Baseline | $17_{\pm 5}$ | $35_{\pm 5}$ | $64_{\pm 10}$ | $102_{\pm 14}$ |
| Trotting ($\boldsymbol{f}^{\text{cmd}} = 2\,\text{Hz}$) | $11_{\pm 2}$ | $25_{\pm 1}$ | $55_{\pm 4}$ | $104_{\pm 8}$ |
| Trotting ($\boldsymbol{f}^{\text{cmd}} = 3\,\text{Hz}$) | $9_{\pm 1}$ | $24_{\pm 1}$ | $53_{\pm 5}$ | $98_{\pm 9}$ |
| Trotting ($\boldsymbol{f}^{\text{cmd}} = 4\,\text{Hz}$) | $9_{\pm 1}$ | $26_{\pm 0}$ | $60_{\pm 4}$ | $114_{\pm 12}$ |
| Trotting ($\boldsymbol{h}_z^{\text{cmd}} = 20\,\text{cm}$) | $9_{\pm 1}$ | $26_{\pm 1}$ | $56_{\pm 3}$ | $102_{\pm 8}$ |
| Trotting ($\boldsymbol{h}_z^{\text{cmd}} = 30\,\text{cm}$) | $9_{\pm 1}$ | $24_{\pm 1}$ | $53_{\pm 5}$ | $98_{\pm 9}$ |
| Trotting ($\boldsymbol{h}_z^{\text{cmd}} = 40\,\text{cm}$) | $10_{\pm 1}$ | $23_{\pm 1}$ | $52_{\pm 4}$ | $95_{\pm 9}$ |

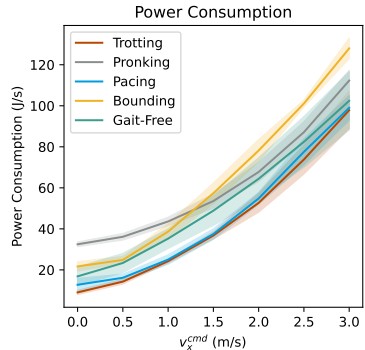

Table 3: Behavior tuning enables interventional studies on the relationship between gait properties and performance criteria within a single policy. Here, we illustrate how power consumption varies across speeds for common quadrupedal gaits and for a baseline policy without gait constraint. Several structured gaits surpass the efficiency of unconstrained gait across all speeds.

**Latency and Actuator Modeling.** Directly identifying invariant properties avoids overly conservative behavior from unnecessary domain randomization. We perform system identification to reduce the sim-to-real gap in the robot dynamics. Following [22], we train an actuator network to capture the non-ideal relationship between PD error and realized torque. Separately, we identify a latency of around $20\,\text{ms}$ in our system and model this as a constant action delay during simulation.

### 3.4 Materials

**Simulator and Learning Algorithm.** We define our training environment in the Isaac Gym simulator [6, 25]. We train policies using Proximal Policy Optimization [26]; details in Appendix A.

**Hardware.** We deploy our controller in the real world on the Unitree Go1 Edu robot [27]. An onboard Jetson TX2 NX computer runs our trained policy. We implement an interface based on Lightweight Communications and Marshalling (LCM) [28] to pass sensor data, motor commands, and joystick state between our code and the low-level control SDK provided by Unitree. For both training and deployment, the control frequency is 50Hz.

**Gait-free Baseline.** To understand the impact of MoB on performance, we compare our controller to a baseline velocity-tracking controller (the "gait-free baseline"). The gait-free baseline is trained by the method above, but excludes all augmented auxiliary rewards (Table 1). Therefore, it only learns one solution to the training environment and its actions are independent of behavior parameters $\mathbf{b}_t$.

## 4 Experimental Results

### 4.1 Sim-to-Real Transfer and Gait Switching

We deploy the controller learned in simulation in the real world and first evaluate its performance on flat ground similar to the training environment. To start, we demonstrate generating and switching between structured gaits that are well-known in the locomotion community. Figure 2 shows torques and contact states during transition between trotting, pronking, bounding, and pacing while alternating $\boldsymbol{f}^{\text{cmd}}$ between $2\,\text{Hz}$ and $4\,\text{Hz}$. We find that all gait parameters are consistently tracked after sim-to-real transfer. Videos (i)-(iv) on the project website visualize the different gaits obtained by modulating each parameter in $\mathbf{b}_t$ individually.

### 4.2 Leveraging MoB for Generalization

**Tuning for New Tasks.** After training using a generic locomotion objective, one might wish to tune a controller's behavior to optimize a new metric in the original environment. MoB facilitates this if some subset of learned behaviors outperform the gait-free policy by the new task metric.

| Gait | $r_{v_{x,y}^{\mathrm{cmd}}}$ | $r_{\omega_z^{\mathrm{cmd}}}$ | $r_{c_f^{\mathrm{cmd}}}$ | $r_{c_v^{\mathrm{cmd}}}$ | Survival |
|---|---|---|---|---|---|
| Trotting | $0.80_{\pm0.01}^{(0.95)}$ | $0.76_{\pm0.00}^{(0.89)}$ | $0.95_{\pm0.00}^{(0.97)}$ | $0.98_{\pm0.00}^{(0.98)}$ | $0.88_{\pm0.01}^{(1.00)}$ |
| Pronking | $0.84_{\pm0.01}^{(0.94)}$ | $0.77_{\pm0.01}^{(0.85)}$ | $0.96_{\pm0.00}^{(0.96)}$ | $0.97_{\pm0.00}^{(0.98)}$ | $0.82_{\pm0.02}^{(1.00)}$ |
| Pacing | $0.76_{\pm0.01}^{(0.91)}$ | $0.76_{\pm0.01}^{(0.81)}$ | $0.94_{\pm0.00}^{(0.96)}$ | $0.98_{\pm0.00}^{(0.98)}$ | $0.87_{\pm0.02}^{(1.00)}$ |
| Bounding | $0.80_{\pm0.01}^{(0.88)}$ | $0.73_{\pm0.01}^{(0.86)}$ | $0.94_{\pm0.00}^{(0.96)}$ | $0.98_{\pm0.00}^{(0.98)}$ | $0.82_{\pm0.01}^{(1.00)}$ |
| Gait-free | $0.81_{\pm0.03}^{(0.96)}$ | $0.74_{\pm0.06}^{(0.92)}$ | – | – | $0.83_{\pm0.01}^{(1.00)}$ |

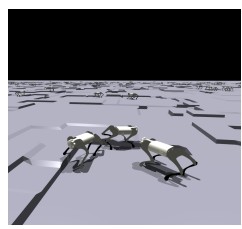

Table 4: Zero-shot generalization to platform terrain (visualized right). Pacing and trotting yield the best survival time in out-of-distribution deployment, outperforming the gait-free baseline. Pronking attains the best velocity tracking performance, with similar survival time to the baseline. We report the fraction of maximum episodic reward. Superscript reports performance in flat training environment with no platforms. Subscript reports standard deviation across three random seeds.

*Energy efficiency (simulated)*: We consider the task of minimizing the mechanical power consumption $(\mathrm{J/s})$ measured by summing the product of joint velocity and torque at each of the 12 motors: $\sum_i \max(\boldsymbol{\tau}_i \dot{\mathbf{q}}_i, 0)$. As shown in Figure 3, several choices of the contact schedule $\boldsymbol{\theta}^{\mathrm{cmd}}$, height $\boldsymbol{h}_z^{\mathrm{cmd}}$, and frequency $\boldsymbol{f}^{\mathrm{cmd}}$ outperform the gait-free policy in this unseen metric, consuming less energy across all speeds. Therefore, tuning the behavior parameters can facilitate tuning performance on a new objective (energy efficiency) in the original training environment.

*Payload manipulation*: We experiment with another task where the robot is required to transport a ball from one place to another, then bend its body so the ball is deposited on the ground. A gait-free policy couldn't do this; while many possible body posture profiles are valid to solve the training environment, the gait-free policy will simply converge to one at random. In a real-world experiment, we demonstrate how MoB-enabled body posture control can be repurposed for teleoperated payload manipulation. The operator pilots the robot to the delivery location with the body level, then pitches the body backward, modulating $\phi^{\mathrm{cmd}}$, to dump the payload (Figure 1, bottom row, second from left).

**Tuning for New Environments.** In the previous section, we showed that MoB can support repurposement to novel tasks in the training environment. In the real world, there is also always a long tail of environments that are not modeled in training. We demonstrate the behavior of our controller trained *only on flat ground* in the presence of challenging non-flat terrains and disturbances such as curbs, bushes, hanging obstacles, and shoves. Successes suggest that the space of policies learned by MoB contains some behaviors that transfer better than the baseline to unseen environments.

*Platform terrain (simulated)*: We evaluate the performance of our robot in traversing randomly positioned platforms with height up to $16\,\mathrm{cm}$ (Table 4). We report two metrics: mean reward and mean survival time as a fraction of the maximum episode length $(10\,\mathrm{s})$. For each metric, we find that by modulating the contact schedule $\boldsymbol{\theta}^{\mathrm{cmd}}$ or footswing height $\boldsymbol{h}_z^{f\,\mathrm{cmd}}$, we can outperform the gait-free policy. (Figure 4, Appendix Figure 9). Therefore, it is possible to improve performance in an out-of-distribution terrain by modulating the parameters of the MoB policy.

*Climbing over curbs*: Previous works demonstrating blind obstacle traversal on a quadruped [1, 2] learned a foot-trapping reflex where the robot first trips, then raises its foot over the obstacle. In contrast, with the help of a human pilot, our gait-conditioned policy with high footswing command enables fast and smooth obstacle traversal without tripping, despite training on a simpler flat-ground terrain. In the real world, we demonstrate that modulating footswing height $\boldsymbol{h}_z^{f\,\mathrm{cmd}}$ enables our controller to climb smoothly across stairs and curbs (Figure 1, bottom left).

*Hacking through thick bushes*: Extremely thick bushes pose a methodological challenge for state-of-the-art perceptive locomotion controllers. They are hard to simulate, because they comply against the whole body of the robot, and they are hard to sense as they are indistinguishable from solid obstacles to depth sensors. Therefore, prior works would either attempt to climb over bushes as obstacles or fall back on a robust proprioceptive controller that is unaware of the semantic context. In contrast, by modulating the gait parameters for gait frequency $\boldsymbol{f}^{\mathrm{cmd}}$ and footswing height $\boldsymbol{h}_z^{f\,\mathrm{cmd}}$ *in situ*, a human operator can guide our system quickly through challenging brush.

*Navigating confined spaces*: Consider the scenario where the robot needs to go under a bar. The gait-free baseline cannot accomplish this; in the absence of such constraints during training, it will

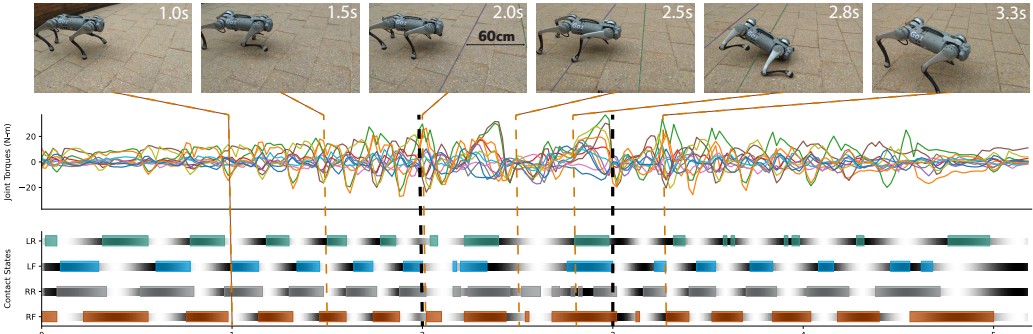

Figure 3: We demonstrate that behavior transitions can be performed even in quick sequence at high speed for synthesis of agile maneuvers. Emulating gap crossing on flat ground, we show that this can facilitate crossing a gap wider than the robot's body length in a single leap.

converge to a fixed body height profile. Our system with MoB can navigate confined spaces through modulation of the body height $h_z^{cmd}$. In a real-world example, the robot was able to crawl under a $22\,cm$ bar; the robot body thickness is $13\,cm$, leaving $9\,cm$ of clearance beneath the robot.

*Anticipative bracing against shoves*: Widening the stance can make a quadruped more robust to hard shoves, but absent a perception module to anticipate a human kick, the robot would always need to walk in a widened stance to be prepared for a shove. This interferes with performance in other tasks like running efficiently, so learned locomotion controllers without MoB often provide incentive to keep the feet nominally below the hips [2, 7]. With MoB, widening the stance width $s_y^{cmd}$ in the pilot's anticipation of a shove enables the robot to remain upright.

**Exploring the Speed and Range of Behavior Adaptation.** If we want to use our low-level controller for high-level tasks, one thing we might desire is to switch between gaits even at high speeds, which might be useful for applications such as parkour. Another is to transition between diverse gaits with precise timing for a synchronous task like dancing.

*Agile forward leap*: As a demonstration of gait transitions at high speed, we modulate contact schedule, velocity, and gait frequency at to encode an agile forward leap (Figure 3). The robot first accelerates to a target speed of $3\,m/s$ at a trot while increasing its step frequency from $2\,Hz$ to $4\,Hz$, then switches to pronking at $2\,Hz$ for one second, then decelerates to a standstill while trotting. During the leap phase, the distance from the location of the robot's front feet at takeoff to the location of the hind feet upon landing is $60\,cm$.

*Choreographed dance*: To demonstrate precisely timed transition between diverse gaits, we program a sequence of gait parameters to generate a dance routine synchronized to a jazz song with a tempo of 90 bpm. At this tempo, combinations of phases 0, 0.25, and 0.5 with frequencies of $1.5\,Hz$ and $3\,Hz$ yield eighth, quarter, half, and full beat gaps between consecutive footsteps. We also modulate body height and velocity in time with the music. An assistant script procedurally generates gait parameters, fed into the controller in open-loop fashion.

## 5    Discussion and Limitations

Our experiments show that the benefits of adding MoB can come at a cost to in-distribution task performance, specifically limiting the robot's flat-ground sprinting performance (Table 5, appendix). Heat maps reveal that our behavior parameterization is restrictive for combinations of high linear and angular velocity. To quantify and control the tradeoff between task performance and reward shaping is an interesting future direction, for which some prior methods have been proposed [29].

MoB confers a single learned policy a structured and controllable space of diverse locomotion behaviors for each state and task in the training distribution. This yields a set of 'knobs' to tune the performance of motor skills in unseen test environments. The system currently requires a human pilot to manually tune its behavior. In the future, the autonomy of our system could be extended by automating behavior selection using imitation from real-world human demonstrations, or by using a hierarchical learning approach to automatically self-tune the controller during deployment.

**Acknowledgments**

We thank the members of the Improbable AI lab for the helpful discussions and feedback on the paper. We are grateful to MIT Supercloud and the Lincoln Laboratory Supercomputing Center for providing HPC resources. This research was supported by the DARPA Machine Common Sense Program, the MIT-IBM Watson AI Lab, and the National Science Foundation under Cooperative Agreement PHY-2019786 (The NSF AI Institute for Artificial Intelligence and Fundamental Interactions, http://iaifi.org/). This research was also sponsored by the United States Air Force Research Laboratory and the United States Air Force Artificial Intelligence Accelerator and was accomplished under Cooperative Agreement Number FA8750-19-2-1000. The views and conclusions contained in this document are those of the authors and should not be interpreted as representing the official policies, either expressed or implied, of the United States Air Force or the U.S. Government. The U.S. Government is authorized to reproduce and distribute reprints for Government purposes, notwithstanding any copyright notation herein.

**Author Contributions**

- **Gabriel B. Margolis** concieved, designed, and implemented the controller, ran all experiments, and played the primary role in paper writing.
- **Pulkit Agrawal** advised the project and contributed to its conceptual development, experimental design, positioning, and writing.

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
