# OpenReview forum: "Walk These Ways: Tuning Robot Control for Generalization with Multiplicity of Behavior"
_robot-learning.org/CoRL/2022/Conference — CoRL 2022 Oral_

### Official Review · Reviewer_fCqU · 2022-07-07

**Originality:** Fair
**Technical Quality:** Good
**Clarity Of Presentation:** Very Good
**Impact:** 3

**Recommendation:**

Weak Reject: I recommend rejecting the paper, but will not argue for my recommendation if the majority of other reviewers have a different opinion.

**Summary:**

This paper presents a framework for learning policies for a quadrupedal robot that are conditioned on different gait parameters. This allows a single policy to generate a diverse set of gaits, including the common trotting, pacing, pronking and bounding. Analysis of the resulting policy shows that different gaits demonstrate different desired characteristics and are used to solve different downstream tasks.

**Issues:**

1. "No previously proposed reinforcement learning agent has been suitable to accomplish any of these tasks", (a) long jump: https://openreview.net/pdf?id=R4E8wTUtxdl, (b) through confined space: no work that I am aware of demonstrates this, but I don't think it is difficult to achieve with existing RL method, e.g., train a pacing gait to go through the narrow space (as is done in this paper), or train a crawling gait with low desired body height. One selling point is that one policy can achieve all these tasks, but the result is not surprising given that all necessary components are already available in the existing literature. Some suggestions, can we extend this framework so that it does less common gaits? e.g., two-legged hopping, skateboarding.

2. In Table 6, similar heat maps for other gaits will be helpful information. What is the commanded stepping frequency for this table? I will imagine high-frequency stepping will be helpful to get better high-speed velocity tracking performance.

3.  In Figure 8, what is the unit of the temperature bar? What is the commanded stepping frequency?

4. Some analysis of how the stepping frequency affects performance can be useful, e.g., a high stepping frequency should be able to achieve faster locomotion as well as more robust to perturbation.

5. I think there are typos in Table 2, e.g., there are two $v_{x}^{cmd}$ in the column.

**Quality Of The Limitations Section:**

Limitations are addressed clearly

**Reviewer Expertise:**

4: The reviewer is confident but not absolutely certain that the evaluation is correct

**Robotics Focus:**

Sufficient demonstration on hardware

**Strengths And Weaknesses:**

Pro:
1. Demonstrate diverse gaits on a quadrupedal robot using a single policy, including successful sim-to-real transfer.

Con:
1. Motion quality degrades significantly when the desired velocity goes up, e.g., the agile jump sequence, compared to what is demonstrated here: https://sites.google.com/view/jumpingfrompixels, https://www.youtube.com/watch?v=9QV0Nnm80tU. Notably, the feet spread wide apart to gain more stability.
2. Some more analysis is needed, see issues below.

**Summary Of Recommendation:**

This paper extends the framework of learning a gait-conditioned policy from Siekmann et al to quadrupedal locomotion. While demonstration on the hardware is appreciated, the sim-to-real success is not surprising given similar demos in the past few years. The system is built upon several prior works with limited algorithmic innovation. Some further analysis of different gaits or more impressive demos that is not achievable by prior work can strengthen the paper.

---

> ### Author Response · Authors · 2022-08-23
> **Response to Reviewer fCqU (1/2)**
>
> We thank the reviewer for their feedback. We have updated our manuscript and provide a detailed response below.
>
> > Motion quality degrades significantly when the desired velocity goes up, e.g., the agile jump sequence, compared to what is demonstrated here: [https://sites.google.com/view/jumpingfrompixels](https://sites.google.com/view/jumpingfrompixels), [https://www.youtube.com/watch?v=9QV0Nnm80tU](https://www.youtube.com/watch?v=9QV0Nnm80tU). Notably, the feet spread wide apart to gain more stability.
>
> Our robot leaps over twice as far at double the speed of the best real-world jump in [https://sites.google.com/view/jumpingfrompixels](https://sites.google.com/view/jumpingfrompixels). It might appear that the motion is degraded because the robot body sinks lower to the ground rather than rising into the air. However, the robot is achieves the task we set out to demonstrate which is to leap across a large horizontal distance. We apologize that the wording “agile jump” may have implied that a vertical jump would occur; there is vertical jumping function in our controller, and we have changed the phrasing to “agile forward leap” in the paper.
>
> One explanation for the dipping body height during the forward leap is that this maneuver involves a rapid drop in gait frequency which makes it difficult to apply forces quickly. In less aggressive maneuvers, where high stepping frequency is maintained, the robot’s high-speed gait transitions can be quite smooth. We have added a new video to illustrate gait switching during running at 3Hz stepping frequency: [https://drive.google.com/file/d/1aVtOckhGsiZaFwDLvswkW4hJZYkSXB15/view?usp=sharing](https://drive.google.com/file/d/1aVtOckhGsiZaFwDLvswkW4hJZYkSXB15/view?usp=sharing). We have also added a note acknowledging this and other scenarios where some performance degradation was observed to the discussion of limitations (Section 5).
>
> > "No previously proposed reinforcement learning agent has been suitable to accomplish any of these tasks", (a) long jump: [https://openreview.net/pdf?id=R4E8wTUtxdl](https://openreview.net/pdf?id=R4E8wTUtxdl),
>
> Our robot leaps over twice as far at double the speed of the best real-world jump in [https://openreview.net/pdf?id=R4E8wTUtxdl](https://openreview.net/pdf?id=R4E8wTUtxdl), indicating that the benefit of our approach can be significant. Where does the benefit come from? This paper ([https://openreview.net/pdf?id=R4E8wTUtxdl](https://openreview.net/pdf?id=R4E8wTUtxdl)) and two related papers from last year’s CoRL ([https://arxiv.org/pdf/2104.04644.pdf](https://arxiv.org/pdf/2104.04644.pdf), [https://openreview.net/pdf?id=NDYbXf-DvwZ](https://openreview.net/pdf?id=NDYbXf-DvwZ)) did not train reinforcement learning agents to execute motor skills. They proposed high-level controllers that learned to modulate the gait of a low-level MPC controller. All used only a low-level MPC controller, because a suitable gait-conditioned RL controller was not available. Our work provides such a gait-conditioned RL controller and would complement these works on task-oriented gait modulation well. It enables the revisitation and extension of these methods with end-to-end learned modules, which we believe would benefit key system bottlenecks in training and deployment.

---

> > ### Comment · Reviewer_fCqU · 2022-08-24
> > **Good additional results. One additional question.**
> >
> > Thanks for the additional results, I believe they make the paper much stronger and I am willing to increase the score.
> >
> > One additional question:
> >
> > The authors keep referencing prior work that claims multi-gait learning is challenging for quadrupedal robots. However, I know one paper that can do multi-gait policy easily, i.e., https://youtu.be/vaivdkRwqAg?t=112, starting at 1:52. Of course, they demonstrate less diverse behaviors and rely on reference motions.
> >
> > It is unclear to me which components of the paper make this possible, i.e., ablations that show if one or more novel and critical parts of the proposed method are removed, then multi-gait learning becomes impossible. Identifying the novel components that make multi-gait learning from challenging to possible can make the contribution of this paper much clearer.

---

> > > ### Author Response · Authors · 2022-08-25
> > > **Followup Response to Reviewer fCqU (1/2)**
> > >
> > > Thank you for your prompt response. We are happy to hear that you find the revised paper much stronger.
> > >
> > > To address your question, we would first like to note the specific training setups where prior work reports that multi-gait training fails or is limited:
> > >
> > > - **Task-guided emergence** [4, 19, 20]: Some works attempt to emerge the gait automatically from an auxiliary reward function, such as the energy penalty in Fu et al [4] and Yang et al [19]. This is reported in these works to be too challenging for end-to-end learning. We suspect this is because, with task reward alone, the policy does not automatically perform enough exploration to discover multiple gaits and select between them.
> > > - **Explicit reference trajectories** [15, 23]**:** Some works (like Xie et al.) provide explicit reference trajectories for each gait directly to the policy. One main limitation here is that extending to a large number of diverse behaviors requires laborious large-scale data generation and labeling. Another is that the policy is not given much freedom to discover useful behaviors that deviate from the reference. For instance, consider the policy we trained without any commands for stance width, body pitch, or footswing height, but still chose these gait properties implicitly for stable locomotion. On the other hand, with a complete reference trajectory, all properties of target gait must be specified by the human, who is not informed of the optimal locomotion behavior before designing, or by an animal in a motion-capture suit, whose optimal gait differs from a robot’s due to morphology differences. Peng et al. 2021 ([https://arxiv.org/pdf/2104.02180.pdf](https://arxiv.org/pdf/2104.02180.pdf)) note: "For many applications, it is not imperative to exactly track a particular reference motion. Since a dataset typically provides only a limited collection of example motions, a character will inevitably need to deviate from the reference motions to effectively perform a given task.” This motivates using a **minimal** set of parameters to specify each gait, which allows our method to optimize for agile and robust locomotion while also expressing diverse gaits.
> > > - **Adversarial motion priors** [5, 6, 7]: AMP is a technique that uses reference motions to impose style without restricting the behavior too much. Peng et al. 2021 ([https://arxiv.org/pdf/2104.02180.pdf](https://arxiv.org/pdf/2104.02180.pdf)) note: “like many other GAN-based techniques, AMP is susceptible to mode collapse. When provided with a large dataset of diverse motion clips, the policy is prone to imitating only a small subset of the example behaviors, ignoring other behaviors that may ultimately be more optimal for a given task.” Although there has been work on extending AMPs to multitask setting [6, 7], it has mostly involved a small number of discrete motion styles on real robots.
> > > - **Gait-conditioned policies** mitigate above mentioned issues. Siekmann et al. reported that additional reward shaping was needed to prevent mode collapse while training with a much smaller set of gaits, but these may be explained by differences between bipeds and quadrupeds, or details of the training setup, and not due to major methodological differences. Our methodology improves on Siekmann et al. in terms of learning gait transition and choice of gait specification, as we describe below.

---

> > > ### Author Response · Authors · 2022-08-25
> > > **Followup Response to Reviewer fCqU (2/2)**
> > >
> > >
> > > The paper from Xie et al provides a nice example of learning residuals for **explicit reference trajectories** in a single policy and we will cite it in our next revision. However, it shares several limitations with other works we cited ([15, 23]):
> > >
> > > 1. Because it relies on penalizing deviation from a fixed reference joint trajectory, the gait parameterization in Xie et al is not scalable and excludes some motions:
> > >     - Reference joint trajectories penalize the robot for developing creative motor strategies that depart from the reference to achieve robustness or agility, but maintain the gait style, such as slip response, perturbation resistance, high speed, etc. As such, Xie et al demonstrate reference-trained gaits moving slowly in a flat laboratory environment (as the focus of their paper is an ablation study of domain randomization)
> > >     - The reference trajectory framework requires the human to do a good job of designing each gait before the robot can learn it, making it infeasible to scale to a very large behavior space. Particularly for compound motions, such as pitching the body while swinging the feet high, it will not be clear to the human designer how the reference trajectory should actually be shaped — this will need to be informed by the physics. Starting with physics-informed trajectory design brings the design process to model-based methods / trajectory optimization, inheriting their characteristics and losing the characteristics of reinforcement learning.
> > >     - Very dynamic gaits like bounding or pronking cannot be specified well by reference joint trajectories. Particularly for high speeds and low frequencies, these gaits are characterized by variable body bouncing and pitching, and consequently their joint trajectories change dramatically with frequency and speed. This is why the pronking in [15], which was based on reference joint trajectory, is very stiff and would not be that useful for tasks like leaping. Similarly, Xie et al only demonstrate the gaits where the body is very still (trotting and pacing).
> > >
> > >     The key contribution of our work that resolves this is our **gait parameterization** that balances between:
> > >
> > >     - being specific enough to produce diverse motions, but not being too restrictive and still allowing flexibility in movement patterns akin to robustness of gait-free methods, and
> > >     - **extensible** to accommodate new gait parameters with little re-tuning, all of which is shown through experiments.
> > >
> > >     Achieving multi-gait policy is possible through:
> > >
> > >     - The specific gait parameterization we propose
> > >     - Randomizing the gait command multiple times within each episode. This exposes the agent to training data that includes gait transitions. We will add a note about this to section 3.2.
> > >
> > >     To specifically answer the reviewers' request for more ablations of the relative importance of these contributions, we are running a few additional experiments. We will share these experiments when available. However, the authors' schedules may not permit completing these experiments before the end of the rebuttal period. If not, we promise to include these results in the next revision of our paper to clarify the impact of each contribution.

---

> > > > ### Comment · Reviewer_fCqU · 2022-08-25
> > > > **Will increase score**
> > > >
> > > > Thanks for the clarification. Now I have a better understanding of the contribution. I will increase the score to accept (looks like the system doesn't allow me to do so for now).

---

> ### Author Response · Authors · 2022-08-23
> **Response to Reviewer fCqU (2/2)**
>
> (continued)
>
>
> >  (b) through confined space: no work that I am aware of demonstrates this, but I don't think it is difficult to achieve with existing RL method, e.g., train a pacing gait to go through the narrow space (as is done in this paper), or train a crawling gait with low desired body height. One selling point is that one policy can achieve all these tasks, but the result is not surprising given that all necessary components are already available in the existing literature.
>
> We agree that training a pacing gait or a crawling gait using RL was within the scope of prior work. The real challenge, as indicated by prior work, is in how to train a policy to achieve many diverse gaits using multi-task learning. Our insight is that training a policy to simultaneously track a many continuous gait parameters enables well-formed, controllable, and practically useful multitask behavior. The other related question is how to parameterize the gaits; we show that a high degree of useful diversity can be achieved by reward shaping, beyond what is achievable by reference trajectories. Our revised manuscript adds three new gait parameters for a total of eleven parameters; please find more information about the extension in our general response. We show that by learning a very large space of gaits in flat-ground locomotion, we can generalize without retraining to performing novel tasks including climbing across obstacles, payload manipulation, navigating constrained environments, leaping, dancing — something prior works didn’t consider.
>
> Surprisingly, one policy can learn to achieve all of these tasks. Prior work has noted the difficulty in learning multiple well-formed gaits in a single policy; take for example Fu et al., 2021 ([https://arxiv.org/pdf/2111.01674.pdf](https://arxiv.org/pdf/2111.01674.pdf)): “Naive Multi-Task Training Fails… We believe the reason for failure is difficulty in optimization as the robot is now tasked not only to learn to move forward but also do it by learning different gaits which causes it to collapse.” Or Siekmann et al., 2021 ([https://arxiv.org/pdf/2011.01387.pdf](https://arxiv.org/pdf/2011.01387.pdf)): “However, learning to transition between gaits by varying both the cycle offsets and phase ratios during training appears to be a challenging learning problem: policies which are trained in this fashion can end up asymmetrically walking instead of hopping, or learn other undesirable behaviors that resemble a fusion of all the different commanded gaits.”
>
> We hope the reviewer will further consider our general response where we provide much more detailed context for the novelty and utility of this work.
>
> > Some suggestions, can we extend this framework so that it does less common gaits? e.g., two-legged hopping, skateboarding.
>
> In agreement with the reviewer’s suggestion, we have added several new gait parameters in our revised manuscript which enable more exotic locomotion styles, including extreme pitch angles, very high footswings, and variable stance width. These additions are described in our general response.
>
> > In Table 6, similar heat maps for other gaits will be helpful information. What is the commanded stepping frequency for this table? I will imagine high-frequency stepping will be helpful to get better high-speed velocity tracking performance. … Some analysis of how the stepping frequency affects performance can be useful, e.g., a high stepping frequency should be able to achieve faster locomotion as well as more robust to perturbation.
>
> Thank you for your suggestions. We have added heatmaps of more quadrupedal gaits to the appendix (Figure 11). The commanded stepping frequency in Table 6 (now Table 5) is 3Hz. As requested, we have added analysis of more gait parameter effects to the appendix (Figure 9): the impact of gait frequency on velocity tracking, and of footswing height on uneven-terrain time to failure. We found that higher gait frequencies yield better tracking performance at high speeds, but 3Hz trotting performs about as well as 4Hz trotting. We also empirically verified that raising the footswing height enables better traversal of the platform terrain.
>
> > In Figure 8, what is the unit of the temperature bar? What is the commanded stepping frequency?
>
> We have added the details: The unit of the temperature bar is unit-normalized reward, and the commanded stepping frequency is 3Hz. We have added these details to the revised manuscript.
>
> > I think there are typos in Table 2, e.g., there are two vxcmd in the column.
>
> Fixed; the second row of the bottom section now correctly reads vycmd.

---

### Official Review · Reviewer_hEjf · 2022-07-22

**Originality:** Good
**Technical Quality:** Excellent
**Clarity Of Presentation:** Excellent
**Impact:** 3

**Recommendation:**

Strong Accept: I recommend accepting the paper and will argue for my recommendation even if other reviewers hold a different opinion.

**Summary:**

This work presents a new RL-based system that learns a single gait-conditioned policy to produce dynamics locomotion movements of a quadruped robot.

**Issues:**

- A clear statement about the differences with [22] are mentioned.
- Table 4 energy measurement explanation.

**Quality Of The Limitations Section:**

Limitations are addressed clearly

**Reviewer Expertise:**

5: The reviewer is absolutely certain that the evaluation is correct and very familiar with the relevant literature

**Robotics Focus:**

Sufficient demonstration on hardware

**Strengths And Weaknesses:**

Strengths:
- The paper is easy to follow and as far as the reviewer is aware of, sound.
- Learning various gaits, with a single policy appears great as an idea.
- The system is implemented on real robots, demonstrating the sim2real challenge.

Weaknesses:
- The robot does not perform well in several cases, i.e., there are instances that it just drags the back feet on the ground, which is not optimal control.
- Is the paper, something more than combining [22], [16], and [25] (and mainly [22])?
- It is very confusing how energy consumption was measured (Table 4)? Some details are needed.

Overall, not much else the reviewer can comment on this work, as far as a clear statement about the differences with [22] are mentioned and of course Table 4 generation. Other than that, this is a really nice paper.

**Summary Of Recommendation:**

If the novelty compared to [22] (and also [16], [25]) is explained and there is indeed some significant work that has been done (other than 2-to-4 legs), it is a nice paper in the field with real-world demonstrations.

---

> ### Author Response · Authors · 2022-08-23
> **Response to Reviewer hEjf**
>
> We thank the reviewer for their feedback. We have updated our manuscript and provide a detailed response below.
>
> > The robot does not perform well in several cases, i.e., there are instances that it just drags the back feet on the ground, which is not optimal control.
>
> We agree that the robot’s behavior is not always optimal. We have expanded the discussion regarding limitations of our system (Section 5) to specifically point out when suboptimal behavior may arise. We now note that constraining the gait may also lead to suboptimality in desiderata besides those we analyzed, such as wear on the robot, footstep noise, or foot dragging. We added that while all gaits we tested were executable in the real world, some suffer from substantial performance degradation. For example, in the agile leap sequence, the robot’s body sinks much lower than the command height during the leap. We also noted that when assuming an unstable gait, such as low-frequency pacing with a narrow stance and high body, the robot tends to drift to one side or stumble.
>
> > Is the paper, something more than combining [22], [16], and [25] (and mainly [22])?
>
> We gladly agree that one main contribution of the paper is synthesizing and analyzing existing methods including [16, 22, 25] into a useful controller for a quadruped. However, our work also extends far beyond the methods of those previous works, demonstrating a simpler and more powerful multitask training methodology which unlocks new capabilities and analysis. Related work including [22] explicitly called for studying highly multitask locomotion policies, expressing uncertainty about their feasibility. The prevailing view was that our result would be challenging to obtain and would not be possible without additional complex methodological innovations. Please further read and consider our general response, in which we provide detailed context supporting the technical novelty of our system. We also describe newly added capabilities which in our opinion further our paper as a significant contribution to the field. We have revised the manuscript to better convey the novelty and include the new experiments.
>
> > It is very confusing how energy consumption was measured (Table 4)? Some details are needed.
>
> As in related works [https://arxiv.org/pdf/2111.01674.pdf](https://arxiv.org/pdf/2111.01674.pdf) and [https://arxiv.org/pdf/2104.04644.pdf](https://arxiv.org/pdf/2104.04644.pdf), we measure power consumption as the sum of torque times angular velocity for each of the 12 motors. The units are kg.m^2/s^2 (torque) x 1/s (angular velocity) = J/s (power). We clip the power consumption for each motor at each timestep as positive, thus assuming there is no energy regeneration. We have added these details to the revised manuscript in section 4.3.

---

### Official Review · Reviewer_E9ME · 2022-07-28

**Originality:** Fair
**Technical Quality:** Very Good
**Clarity Of Presentation:** Excellent
**Impact:** 3

**Recommendation:**

Weak Reject: I recommend rejecting the paper, but will not argue for my recommendation if the majority of other reviewers have a different opinion.

**Summary:**

In this paper, the authors propose a new system to train diverse locomotion skills in a single policy. Compared with previous works where only the moving speeds and angular speeds are tracked, the proposed policy architecture also takes into account the preferred gait parameters such as frequency and phase offsets. With a curriculum training procedure, the authors enable the quadruped robot to perform trotting, pacing, pronking, and bounding, all using a single learned policy. The authors zero-shot deploy the policy to a real go-1 robot and show that the gait conditioned policy’s performance is better than gait-free baselines.


**Issues:**


Other issues of the paper, in no particular order:

1, In Table 2, the notation v_x^cmd appears twice.

2, In Table 4, prefer to use the cost of transport over power consumption, since CoT is more standard in locomotion literature.

3, There are also works related to gait transition and learning such as: Fast and Efficient Locomotion via Learned Gait Transition, which is not discussed in the related works.

4, What is the control frequency of the policy?

5, The gait-free baseline. “The pacing and trotting gaits yield the best survival time during zero-shot deployment on this particular terrain, outperforming the gait-free baseline.” This is a bit surprising. Since the gait free baseline with auxiliary rewards pretty much will convert to trotting, and thus I would expect similar (good) stability.

6, In Figure 4, please mark the gait transitions on the gait pattern subfigure to help visualization.  Also, mark the timing of each robot figure on top.

7, The “agile jumping” is hardly a successful jump as I can tell from the video. I think during the training there is no reward to incentivize the jumping height. I would only call this  as a bounding gait with a small aero phase.


**Quality Of The Limitations Section:**

Limitations are addressed clearly

**Reviewer Expertise:**

5: The reviewer is absolutely certain that the evaluation is correct and very familiar with the relevant literature

**Robotics Focus:**

Sufficient demonstration on hardware

**Strengths And Weaknesses:**

The strengths of this paper:

The paper is clearly written and easy to follow. The results are comprehensive and the hardware evaluations are rich. The authors measure various aspects including stability and power consumption.

The weaknesses of this paper:

1, Limited novelty. The delta between this and previous works seem to be mainly the inclusion of gait parameters during the RL training. All other aspects, such as policy architecture, reward shaping, and training curriculums, seem to be a reimplementation of previous works.

2, Limited baselines. There is only a gait-free baseline that the author is compared with. I think it will be great to include other baselines such as these trained with motion imitation methods.




**Summary Of Recommendation:**

The paper seems to have limited novelty and little comparisons with enough baselines.

---

> ### Author Response · Authors · 2022-08-23
> **Response to Reviewer E9ME (1/2)**
>
> We thank the reviewer for their feedback. We have updated our manuscript and provide a detailed response below.
>
> > 1, Limited novelty. The delta between this and previous works seem to be mainly the inclusion of gait parameters during the RL training. All other aspects, such as policy architecture, reward shaping, and training curriculums, seem to be a reimplementation of previous works.
>
> Our method learns gaits that are much more diverse than previous methods. We further show these gaits to enable many practically useful behaviors and generalization to new applications from a single system. An update to our manuscript includes newly added gait parameters which increase the delta in capabilities from previous work. Please read and consider our general response where we expand on the differences between our work and the approaches in the existing literature.
>
> > 2, Limited baselines. There is only a gait-free baseline that the author is compared with. I think it will be great to include other baselines such as these trained with motion imitation methods.
>
> Gait-conditioned policies provide an automatic way to learning diverse skills without relying on additional supervision or data requirements of motion libraries. In comparison to motion imitation, our work (i) mitigates the need for manual data collection and generation / labeling; (ii) produces gaits that would not be present in motion capture datasets — for example, extremely high footswings or severe body pitch which are not common in animals; and (iii) the gait-conditioned skills are complementary to those learned by motion imitation; future works can consider combining them other. There could be some scenarios where motion imitation is easier than RL, or the better choice for a particular task. However, this choice is problem-dependent, and our contribution is in expanding the space of useful skills we can learn in a single policy using RL.
>
> In this context, our existing gait-free baseline, which excludes gait parameters, is a strong and commonly used baseline approach for RL-based locomotion controllers. It is shown to be proficient at high-speed running, but cannot perform many of the tasks we demonstrate with our method in a single controller, such as climbing quickly across platforms, ducking under obstacles, leaping, or dancing. This illustrates the benefit of adding gait parameters.
>
> > In Table 2, the notation v_x^cmd appears twice.
>
> Fixed in Table 6 (formerly Table 2) of the revision; the second row now correctly reads v_y^cmd
>
> > In Table 4, prefer to use the cost of transport over power consumption, since CoT is more standard in locomotion literature.
>
> We would prefer to use power consumption since CoT can have ambiguous meaning: it may include only applied mechanical power, or may also include energy loss in the motors and energy consumed by computing, which are difficult to model in simulation. This might result in implying a fair comparison across studies, while in practice mechanical differences in the system, physics simulator, etc may be critical. Instead, we find it clearer to measure power consumption here since our goal is only to compare the performance among our method’s different gaits and against baselines. We have added the exact method used to compute power consumption in the paper.
>
> > There are also works related to gait transition and learning such as: Fast and Efficient Locomotion via Learned Gait Transition, which is not discussed in the related works.
>
> We have added this reference and another on learning gait transition in the related works. At CoRL 2021, this paper and two related papers ([https://arxiv.org/pdf/2110.15344.pdf](https://arxiv.org/pdf/2110.15344.pdf), [https://openreview.net/pdf?id=NDYbXf-DvwZ](https://openreview.net/pdf?id=NDYbXf-DvwZ)) proposed high-level controllers that learned to modulate gait for energy efficiency or visual locomotion. All used a low-level MPC controller, because a suitable gait-conditioned RL controller was not available. Our work provides a suitable controller and invites the revisitation and extension of these methods with end-to-end learned modules, which could resolve key system bottlenecks.

---

> ### Author Response · Authors · 2022-08-23
> **Response to Reviewer E9ME (2/2)**
>
> (continued)
>
> > What is the control frequency of the policy?
>
> 50Hz; added to the revised manuscript (line 151).
>
> > The gait-free baseline. “The pacing and trotting gaits yield the best survival time during zero-shot deployment on this particular terrain, outperforming the gait-free baseline.” This is a bit surprising. Since the gait free baseline with auxiliary rewards pretty much will convert to trotting, and thus I would expect similar (good) stability.
>
> We agree that this is an interesting result, and we think it makes sense considering that the platform terrain is outside the policy’s training distribution. Of course, if retrained on platformed terrain, with task-specific tuning of the terrain difficulty, layout, and reward, the gait-free policy could converge to a robust gait on platforms. But this is not our goal:
>
> In any machine learning system, we train in some distribution and would like our system to perform well when tested on a new distribution. No matter what training distribution is used, we may always encounter some out-of-distribution task in the real world. Our goal in learning diverse gaits is to build a system that can be adapted to new tasks without retraining. To make this comparison stark, we only train on flat ground but aim to synthesize behaviors diverse enough that some of them work on other tasks and terrains.
>
> Based on our analysis, when trained on flat ground, a gait-free policy finds a single “hyper-specialized” for flat ground traversal for different speeds. On the other hand, gait-conditioned policies learn multiple walking strategies at each speed, and if these are substantially diverse, each will have different performance in a new environment, with potentially some better than the gait-free policy and some worse. This is why we think gait-conditioned policies are interesting: they enable searching for a desirable gait in a new unseen environment, without retraining the policy.
>
> > In Figure 4, please mark the gait transitions on the gait pattern subfigure to help visualization. Also, mark the timing of each robot figure on top.
>
> Thank you for the suggestion. We have applied this change to improve the figure.
>
> > The “agile jumping” is hardly a successful jump as I can tell from the video. I think during the training there is no reward to incentivize the jumping height. I would only call this as a bounding gait with a small aero phase.
>
> We are sorry for the confusion: we recognize that our choice of words might have made the reader expect a vertical jump. To avoid a false impression, we have modified the paper to refer to “forward leap” rather than “jump".
>
> Our robot leaps over twice as far at double the speed of the best real-world jump in [https://sites.google.com/view/jumpingfrompixels](https://sites.google.com/view/jumpingfrompixels). It might appear that the motion is degraded because the robot body sinks lower to the ground rather than rising into the air. However, the robot is achieves the task we set out to demonstrate which is to leap across a large horizontal distance. Our controller would be useful, for example, for revisiting the Jumping from Pixels work to potentially cross larger gaps at higher speeds.
>
> One explanation for the dipping body height during the forward leap is that this maneuver involves a rapid drop in gait frequency which makes it difficult to apply forces quickly. In less aggressive maneuvers, where high stepping frequency is maintained, the robot’s high-speed gait transitions can be quite smooth. We have added a new video to illustrate gait switching during running at 3Hz stepping frequency: [https://drive.google.com/file/d/1aVtOckhGsiZaFwDLvswkW4hJZYkSXB15/view?usp=sharing](https://drive.google.com/file/d/1aVtOckhGsiZaFwDLvswkW4hJZYkSXB15/view?usp=sharing). We have also added a note on this and other scenarios where some performance degradation was observed to the discussion of limitations (Section 5).

---

### Official Review · Reviewer_kWnF · 2022-07-30

**Originality:** Very Good
**Technical Quality:** Good
**Clarity Of Presentation:** Very Good
**Impact:** 4

**Recommendation:**

Weak Accept: I recommend accepting the paper, but will not argue for my recommendation if the majority of other reviewers have a different opinion.

**Summary:**

This paper presents a gait controller for quadruped robots where the controller is parameterized by target velocities, feet patterns, stepping frequency, and the trunk height. So the controller can satisfy user-provided target velocity commands while generating various gait styles, which enables users to easily adapt their commands to achieve the desired tasks. More specifically, the controller is learned by deep RL where it is asked for achieving a random command generated by a sampling scheme that is adaptive to the current controller performance. Additionally, domain randomization is included for sim-to-real transfer.


**Issues:**

Please see “Suggestion” above

**Quality Of The Limitations Section:**

Additional details required

**Reviewer Expertise:**

3: The reviewer is fairly confident that the evaluation is correct

**Robotics Focus:**

Sufficient demonstration on hardware

**Strengths And Weaknesses:**

Strength

> The learned controller is robust and shows impressive sim-to-real performance, so it can be deployed in many challenging environments. The gait styles that the controller generates are diverse, which also include some agile motions. By considering these two factors, it has potential to be used as a base controller in many quadruped robot platforms.

> The paper is well-written and the exposition is mostly clear.


Weakness

> Not much of technical novelties were presented, where most components came from the existing methods.

> No failure case is explicitly mentioned in neither the paper nor the supplemental material.


Suggestion

> Why are the contact sensor inputs not given to the controller? Is there any particular reason for this? Because the command and the reward function requires specific foot patterns, I guess such information would be really helpful for better performance.

> Figure3: it would be better if the horizontal axis could be” # of simulation step” instead of “training iteration” so that readers know how many samples would be necessary for the training.

> The motions generated do not seem energy efficient, can we improve this by adding more weights on it?

> Table6: the figure on the right is not symmetric w.r.t. w_z^{cmd}. Although I don’t think it should be perfectly symmetric due to randomness in nature, it should be approximately symmetric because there exists the adaptive sampling algorithm and the robot is perfectly symmetric at least in the simulation. For example, there is a significant performance difference between (-2.5, -2.5) and (-2.5, 2.5).

> “One might hypothesize that the most energy-efficient gaits are also the easiest to learn, but our results go against this hypothesis (pronking emerges earlier than pacing; see Section 4.1).” -> This might also be due to the balancing problem where pronking or bounding has less balancing issue than the others.

> I don’t find the definition for {\hat q} in Table1. Is it the previous joint positions?


**Summary Of Recommendation:**

Although there’s not many novel technical components, finding the right combination that would work well in the real world is not a trivial problem. I think that the combination presented in this paper is simple but very powerful enough to be used as a base controller in many existing quadruped platforms.

---

> ### Author Response · Authors · 2022-08-23
> **Response to Reviewer kWnF**
>
> We thank the reviewer for their feedback. We have updated our manuscript and provide a detailed response below.
>
> > Not much of technical novelties were presented, where most components came from the existing methods.
>
> Please read and consider our general response where we add motivation for the novelty of our approach. We also describe newly added capabilities that increase the delta from previous works.
>
> > No failure case is explicitly mentioned in neither the paper nor the supplemental material.
>
> We appreciate the suggestion to elaborate on failure cases. We have expanded the limitations section (section 5) in the revised manuscript. We added that while all gaits we tested were executable in the real world, some suffer from substantial performance degradation. For example, in the agile leap sequence, the robot’s body sinks much lower than the command height during the leap. We also noted that when assuming an unstable gait, such as low-frequency pacing with a narrow stance and high body, the robot tends to drift to one side or stumble.
>
> We also note that Table 6 (Now Table 5, ”Removing gait constraints results in an improvement in velocity tracking task performance”) can be considered to illustrate a failure case in detail: gait-conditioned policies do not achieve as high speed on flat ground as the gait-free baseline. However, this is not surprising — generalization comes at a small cost in performance on the training environment.
>
> > It has potential to be used as a base controller in many quadruped robot platforms.
>
> We plan to fully open-source our training and deployment code upon publication so that this benefit of our work can be maximized. More details are provided in the general response.
>
> > Why are the contact sensor inputs not given to the controller? Is there any particular reason for this? Because the command and the reward function requires specific foot patterns, I guess such information would be really helpful for better performance.
>
> We tested this and did not find any meaningful performance improvement. The observation history likely already contains enough information to infer the contact state. Because we achieve high performance without observing the contact state, this controller has the benefit that it can also be deployed on robots without contact sensors. We have added a note on line 154 of the revision.
>
> > Figure3: it would be better if the horizontal axis could be” # of simulation step” instead of “training iteration” so that readers know how many samples would be necessary for the training.
>
> We appreciate the suggestion. The axis has been changed in the revised paper.
>
> > The motions generated do not seem energy efficient, can we improve this by adding more weights on it?
>
> Our paper is focused on generating diverse gaits. As we show, some are more or less energy efficient due to the constraints on their motion. In the future, work like [https://arxiv.org/pdf/2104.04644.pdf](https://arxiv.org/pdf/2104.04644.pdf) could be directly applied as a high-level policy to modulate our policy’s gait for energy efficiency.
>
> > Table6: the figure on the right is not symmetric w.r.t. w_z^{cmd}. Although I don’t think it should be perfectly symmetric due to randomness in nature, it should be approximately symmetric because there exists the adaptive sampling algorithm and the robot is perfectly symmetric at least in the simulation. For example, there is a significant performance difference between (-2.5, -2.5) and (-2.5, 2.5).
>
> The figure on the right shows the gait-free policy which has no explicit incentive for symmetry. One recent work studied the symmetry of learned locomotion in detail: [https://arxiv.org/pdf/2207.00797.pdf](https://arxiv.org/pdf/2207.00797.pdf) and found that a “data symmetry loss” phenomenon can lead to asymmetric policy performance in high-speed running. Since the asymmetry did not seem too strong in our setting, we opted not to include their proposed “mirror-world network” strategy. But, it could be useful if the asymmetry is substantially impacting performance for some applications.
>
> > “One might hypothesize that the most energy-efficient gaits are also the easiest to learn, but our results go against this hypothesis (pronking emerges earlier than pacing; see Section 4.1).” -> This might also be due to the balancing problem where pronking or bounding has less balancing issue than the others.
>
> Agreed. We ended up removing our original statement for space but we think it is an interesting point for thought. Please let us know if there is anything on this topic you would find especially important to include in the paper.
>
> > I don’t find the definition for {\hat q} in Table1. Is it the previous joint positions?
>
> {\hat q} are the nominal joint positions (standing pose). The definition has been added in revision line 120-121.

---

> > ### Comment · Reviewer_kWnF · 2022-08-24
> > **Thanks for the response.**
> >
> > Thanks for the response. The rebuttal resolved my concerns by clarifying technical novelties, and new results are promising as well. So, I would increase my score to "Accept".

---

> > > ### Author Response · Authors · 2022-08-25
> > > **Followup Response to Reviewer kWnF**
> > >
> > > Thank you, we are glad that our response resolved your concerns and that you find the new results promising.

---

### Author Response · Authors · 2022-08-23
**General Response**

We thank the reviewers for their constructive and thoughtful feedback. All reviewers agreed that the paper is presented clearly, of high technical quality, and that diverse quadrupedal gaits learned in a single policy were comprehensively demonstrated to be useful for downstream tasks through real world experiments. However, some reviewers express the belief that our result follows obviously from recent works, or that it has low novelty relative to reference [22] (which has changed to [24] in the revised manuscript). We provide a general response here and respond to each reviewer’s detailed questions separately.

First, we describe the state of learning-based locomotion today. Most learned locomotion controllers are optimized for one task and typically they converge to a single motion style. To produce a generalist locomotion agent that can complete multiple tasks, we want our controller to produce multiple gaits and postures, which prior work had found challenging to implement in a single policy. Our core innovations include:

1. *Gait-conditioned policies*, a mechanism for learning multiple gaits in a single policy.
2. We demonstrate a multi-gait policy can enable the robot to perform tasks it was not explicitly trained for (i.e., *generalization*).

Our claims are further supported by **new results with more behaviors:** We add new experiments wherein we train a policy with 11 gait parameters instead of 8 at the time of submission. No modification of any existing reward terms or weighting of reward terms was required, which demonstrates that no careful reward tuning is required to scale our method to more gait parameters. Details of these new behaviors are provided below.

As described in more detail below, it is well known that learning a single policy for multiple gaits is a challenging problem. We propose technical ideas to overcome this problem that are inspired by [22]. However, quoting from [22] “**Some questions which remain unanswered include how easily this framework might be applied to the space of possible quadrupedal gaits**”. Our work overcomes this non-trivial challenge posed in [22]. Furthermore, in comparison to [22]:

- [22] only demonstrated a very limited space of gaits and did not show that the gaits can be composed together to solve downstream tasks.
- Composing gaits to solve downstream tasks leads to another challenge — learning to  transition between gaits. As [22] says: “However, learning to transition between gaits by varying both the cycle offsets and phase ratios during training **appears to be a challenging learning problem**: policies which are trained in this fashion can end up asymmetrically walking instead of hopping, or learn other undesirable behaviors that resemble a fusion of all the different commanded gaits.” We show that our training paradigm overcomes this significant problem.

Therefore not only is our work overcoming technical challenges posed in [22] and other prior work, but also leading to new capabilities: online application to new downstream tasks and using a single policy to solve multiple tasks which were previously thought to be hard problems.

**Open Source Intent:** Finally, because our controller allows composing varied gaits to accomplish different tasks, we hope it will useful tool for future locomotion research. We promise to support the research community by open-sourcing our code. This will include complete code for training, evaluation, and deployment with the Unitree SDK.

It required substantial effort to reimplement and integrate state-of-the-art components into a strong generalist low-level locomotion policy. We hope it can serve as a base to catalyze future work on developing high-level controllers for specific tasks.

Below we expand upon each of these points in more detail.

---

> ### Author Response · Authors · 2022-08-23
> **New Results with More Behaviors**
>
> Our work is a demonstration that a single locomotion policy can embody a large number of parameterized gaits useful for zero-shot application to downstream tasks. To strengthen this point, we add three new controllable gait parameters to our formulation: footswing height, body pitch, and stance width. We demonstrate ([https://sites.google.com/view/gait-conditioned-rl/](https://sites.google.com/view/gait-conditioned-rl/)) that these enable novel capabilities for a blind robot:
>
> - **Footswing height control** lets the robot quickly climb over platform obstacles without tripping. ([https://drive.google.com/file/d/1EwkA7XGTgM3xdZiwmVb43d6Onm4RFW-l/view?usp=sharing](https://drive.google.com/file/d/1EwkA7XGTgM3xdZiwmVb43d6Onm4RFW-l/view?usp=sharing))
> - **Body pitch control** lets the robot manipulate objects by pushing them or tilting off a mounted payload. It would also be useful for controlling the angle of a mounted camera. ([https://drive.google.com/file/d/1hYTX-l4vLysUAcomAifgLhPKmO-JErOt/view?usp=sharing](https://drive.google.com/file/d/1hYTX-l4vLysUAcomAifgLhPKmO-JErOt/view?usp=sharing))
> - **Stance width modulation** is shown to influence body stability during large pushes. The robot is less agile in a very wide stance but its body rolls much less under perturbation. ([https://drive.google.com/file/d/1R8CT9nnDgpv3EwkBZtYJmH6HPFq4H_AQ/view?usp=sharing](https://drive.google.com/file/d/1R8CT9nnDgpv3EwkBZtYJmH6HPFq4H_AQ/view?usp=sharing))
>
> The AC mentions that “the weights of the reward terms may need to be carefully chosen for successful implementation.” We report with our new result that the original reward terms and weights do not need any modification or re-tuning to add these new gait parameters and capabilities using our proposed framework.
>
> Our expanded system has eleven independently controllable continuous gait parameters which can be changed in any combination. In combination, these parameters encode thousands of visibly distinct gaits. We did not find any parameter setting in our experiments to result in sim-to-real failure on flat ground, although some limitations are elaborated on in section 5 of the revision. The various gaits are also useful for performing interesting and practical tasks. In contrast, prior works have not shown that a single policy can learn a large corpus of diverse and useful skills. Next, we provide some context on why this result is surprising and valuable to the community.

---

> ### Author Response · Authors · 2022-08-23
> **Novelty and Significance (1/2)**
>
> Our work shows that making policies gait-conditioned yields extremely diverse quadrupedal gaits in one small, fully-connected policy network. We both introduce gait conditioning as a  framework and provide a novel method for its implementation. The problem of how to make locomotion multi-task is of significant, long-standing interest to researchers in the field, who often noted this explicitly in calls for future work:
>
> - Hwangbo et al., Science Robotics 2019: “A single neural network trained in one session manifests single-faceted behaviors that do not generalize across multiple tasks. Introducing hierarchical structure in the policy network can remedy this and is a promising avenue for future work.”
> - Lee et al., Science Robotics 2020: “We see a number of limitations and opportunities for future work. First, the presented controller only exhibits the trot gait. This is narrower than the range of gait patterns discovered by quadrupeds in nature. The gait pattern is constrained in part by the kinematics and dynamics of the robot, but the ANYmal machines are physically capable of multiple gaits. We hypothesize that training protocols and objectives that emphasize diversity can elicit these.”
> - Yang et al., Science Robotics 2020: “While scaling up the number of locomotion modes, training in physics simulation may impose some limitations. Though all policies were validated by the Jueying robot in five locomotion modes, the discrepancy between the simulation and the real world may accumulate and arise as an issue, when the number of tasks increases.”
>
> While there has been long-standing community interest in learning diverse quadrupedal gaits end-to-end, several excellent efforts attempted this task and reported problems, as recently as last year’s CoRL:
>
> - Yang et al., CoRL 2021 ([https://arxiv.org/pdf/2104.04644.pdf](https://arxiv.org/pdf/2104.04644.pdf)): “We find that E2E [end-to-end RL] only learns to stand, while PMTG learns an unnatural gait of using 3 legs, and cannot track the desired speed well.”
> - Fu et al., CoRL 2021 ([https://arxiv.org/pdf/2111.01674.pdf](https://arxiv.org/pdf/2111.01674.pdf)): “Naive Multi-Task Training Fails… We believe the reason for failure is difficulty in optimization as the robot is now tasked not only to learn to move forward but also do it by learning different gaits which causes it to collapse.”
>
> Our work overcomes these problems by proposing gait-conditioned locomotion policies. Siekmann et al. [22] inspired our approach by learning a family of bipedal gaits. However, the space of behaviors learned in [22] is (a) much smaller and (b) has manually specified auxiliary rewards for specific gaits. The multi-gait policy in [22] has just two parameters, the cycle offsets (timing offset between the two feet) and the phase ratios (duration of the swing phase). The combination of many interacting gait parameters including body posture control, footswing control, and our richer space of contact schedules is not demonstrated. Additionally, the authors report the challenges of eliciting multiple gaits from one policy:
>
> - [22]:  “However, learning to transition between gaits by varying both the cycle offsets and phase ratios during training **appears to be a challenging learning problem**: policies which are trained in this fashion can end up asymmetrically walking instead of hopping, or learn other undesirable behaviors that resemble a fusion of all the different commanded gaits.”
>
> The authors then implement additional auxiliary reward shaping to achieve specific gaits, unlike in our result. Subsequently, the authors of [22] explicitly pose our research question as future work with uncertainty about its success:
>
> - [22]: “**Some questions which remain unanswered include how easily this framework might be applied to the space of possible quadrupedal gaits**”.
>
> We answer this question with the gait-conditioned framework which more than extending [22] to quadrupeds, expands it to include a large space of eleven continuous gait parameters for different aspects of the robot’s motion. We present an implementation that critically learns not only to execute diverse gaits, but also to transition between them during deployment. This enables them to be modulated in real time to produce diverse behaviors useful for diverse downstream tasks — none of which was proposed or demonstrated in prior work.
>
> (continued)

---

> ### Author Response · Authors · 2022-08-23
> **Novelty and Significance (2/2)**
>
> (continued)
>
> Our framework and implementation also offer immediate benefit to a popular line of work in recent years on high-level gait modulation. At CoRL 2021, three papers ([https://arxiv.org/pdf/2104.04644.pdf](https://arxiv.org/pdf/2104.04644.pdf), [https://arxiv.org/pdf/2110.15344.pdf](https://arxiv.org/pdf/2110.15344.pdf), [https://openreview.net/pdf?id=NDYbXf-DvwZ](https://openreview.net/pdf?id=NDYbXf-DvwZ)) proposed high-level controllers that learned to modulate gait for energy efficiency or visual locomotion. All used a low-level MPC controller, because a suitable gait-conditioned RL controller was not available. Our work provides a suitable controller and invites the revisitation and extension of these methods with end-to-end learned modules, which would resolve key system bottlenecks. We hope that open-sourcing our code will catalyze research in this area.

---

### Author Response · Authors · 2022-08-23
**Revised Manuscript and Appendix**

**Comment:**

Sections revised in response to reviewer comments are in red. The revised appendix is contained in the attached zip file. Please also see our project website for new videos: https://sites.google.com/view/gait-conditioned-rl/home

**Zip File:**

/attachment/37b2475da35e56476acc86f907bd03b8a4c76ec4.zip

---

### Author Response · Authors · 2022-08-28
**Concluding the Rebuttal Period**

We would like to thank all participants for their time and their contributions to our paper during the review and rebuttal period. We were able to confirm that we favorably addressed the inquiries of 3/4 reviewers during this time. These reviewers also expressed that the new additional experiments enhance the paper's quality and impact. We did not receive a follow-up response from reviewer E9ME, so we do not know if they have further concerns, but we have done our best to address them. If they have time to respond later, we would also love to incorporate any further suggestions they may have for this paper.

---

### Meta-Review · Area_Chair_QMbV · 2022-08-08

**Recommendation:** Accept (Oral)
**Confidence:** 4

**Metareview:**

### Initial Meta-review:

#### Strength:

The paper is clearly written and easy to follow. Successful real-world demonstration of the learned single policy controller on a physical quadruped robot is presented over different gait styles and on challenging terrains. Detailed evaluations on various aspects including robustness and energy consumption are provided.

#### Weakness:

One of the weaknesses of this paper is lack of novelty. The proposed method appears to be a combination of existing methods. In addition, the weights of the reward terms may need to be carefully chosen for successful implementation. It would be useful to include comparison to other baselines.

### Final Meta-review:

The authors have sufficiently addressed the comments by the reviewers with additional results, and clarified the novelty and significance. Majority of the reviewers agree that this paper be accepted.

**Best Paper Nomination:**

No

---

> ### Author Response · Authors · 2022-08-23
> **Response to Area Chair QMbV**
>
> We thank the AC and the reviewers for their thoughtful feedback. We have revised the manuscript and responded to the initial reviews. Please refer to our general response for the following which address your concerns:
> - Detailed context on the novelty and significance of our work, and its relation to other methods/baselines.
> - New experiments demonstrating the extensibility of our work to more tasks and parameters without any re-tuning of the original reward terms.
>
> (General response: https://openreview.net/forum?id=52c5e73SlS2&noteId=GVrifkwgHs)